# Log-Likelihood Ratio Minimizing Flows:
# Towards Robust and Quantifiable
# Neural Distribution Alignment

**Ben Usman** [1,2]
usmn@bu.edu

**Avneesh Sud** [2]
asud@google.com

**Nick Dufour** [2]
ndufour@google.com

**Kate Saenko** [1,3]
saenko@bu.edu

Boston University [1]    Google AI [2]    MIT-IBM Watson AI Lab [3]

## Abstract

Distribution alignment has many applications in deep learning, including domain adaptation and unsupervised image-to-image translation. Most prior work on unsupervised distribution alignment relies either on minimizing simple non-parametric statistical distances such as maximum mean discrepancy or on adversarial alignment. However, the former fails to capture the structure of complex real-world distributions, while the latter is difficult to train and does not provide any universal convergence guarantees or automatic quantitative validation procedures. In this paper, we propose a new distribution alignment method based on a log-likelihood ratio statistic and normalizing flows. We show that, under certain assumptions, this combination yields a deep neural likelihood-based minimization objective that attains a known lower bound upon convergence. We experimentally verify that minimizing the resulting objective results in domain alignment that preserves the local structure of input domains.

## 1   Introduction

The goal of unsupervised domain alignment is to find a transformation of one dataset that makes it similar to another dataset while preserving the structure of the original. The majority of modern neural approaches to domain alignment directly search for a transformation of the dataset that minimizes an empirical estimate of some statistical distance - a non-negative quantity that takes lower values as datasets become more similar. The variability of what "similar" means in this context, which transformations are allowed, and whether data points themselves or their feature representations are aligned, leads to a variety of domain alignment methods. Unfortunately, existing estimators of statistical distances either restrict the notion of similarity to enable closed-form estimation [24], or rely on adversarial (min-max) training [27] that makes it very difficult to quantitatively reason about the performance of such methods [3; 5; 25]. In particular, the value of the optimized adversarial objective conveys very little about the quality of the alignment, which makes it difficult to perform automatic model selection on a new dataset pair. On the other hand, Normalizing Flows [20] are an emerging class of deep neural density models that do not rely on adversarial training. They model a given dataset as a random variable with a simple known distribution transformed by an unknown invertible transformation parameterized using a deep neural network. Recent work on normalizing flows for maximum likelihood density estimation made great strides in defining new rich parameterizations for these invertible transforms [8; 11; 14], but little work focused on flow-based density alignment [12; 29].

In this paper, we present the Log-likelihood Ratio Minimizing Flow (LRMF), a new non-adversarial approach for aligning distributions in a way that makes them indistinguishable for a given family of density models $M$. We consider datasets $A$ and $B$ indistinguishable with respect to the family $M$ if there is a single density model in $M$ that is optimal for both $A$ and $B$ individually since in this case

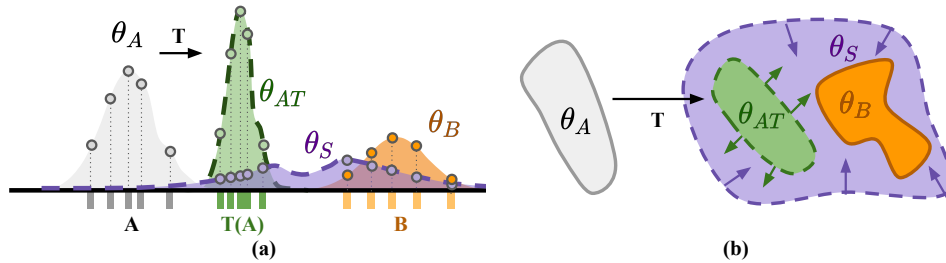

**Figure 1:** To align input datasets $A$ and $B$, we look for a transformation $T$ that makes $T(A)$ and $B$ "indistinguishable". **(a)** We propose the log-likelihood ratio distance $d_\Lambda(T(A), B)$ that compares likelihoods of density models $\theta_{AT}$ fitted to $T(A)$ and $\theta_B$ to $B$ independently with the likelihood of $\theta_S$ optimal for the combined dataset $T(A) \cup B$. This problem is adversarial, but we show how to reduce it to minimization if $T$ is a normalizing flow. **(b)** Colored contours represent level sets of models for $B$ (orange), $T(A)$ (green), and $\theta_S$ (purple), contour sizes corresponds to entropies of these models. Only $\theta_{AT}$ and $\theta_S$ (dashed) change during training. The proposed objective can be viewed as maximizing the entropy of the transformed dataset, while minimizing the combined entropy of $T(A) \cup B$, i.e. expanding green contour while squeezing the purple contour around green and orange contours. At equilibrium, $\theta_S$ and $\theta_{AT}$ model the same distribution as $\theta_B$, i.e. shapes of purple and green contours match and tightly envelope the orange contour. *Best viewed in color.*

there is no way of telling which of two datasets was used for training it. For example, two different distributions with the same means and covariances are indistinguishable for the Gaussian family $M$ since we can not tell which of two datasets was used by examining the model fitted to either one of them. For a general $M$, we can quantitatively measure whether two datasets are indistinguishable by models from $M$ by comparing average log-likelihoods of two "private" density models each fit independently to $A$ and $B$, to the average log-likelihood of the "shared" model fit to both datasets at the same time. We observe that, if datasets are sufficiently large, the maximum likelihood of the "shared" model would reach the likelihoods of two "private" models on respective datasets only if the shared model is optimal for both of them individually, and consequently datasets are equivalent with respect to $M$. Then a density model optimal for $A$ is guaranteed to be optimal for $B$ and vice versa.

We want to find a transformation $T(x)$ that transforms dataset $A$ in a way that makes the transformed dataset $T(A)$ equivalent to $B$ for the given family $M$. We do that by minimizing the aforementioned gap between average log-likelihood scores of "shared" and "private" models. In this paper, we show that, generally, such $T(x)$ can be found only by solving a min-max optimization problem, but if $T(x, \phi)$ is a family of normalizing flows, then the flow $T(x, \phi^*)$ that makes $T(A, \phi^*)$ and $B$ equivalent with respect to $M$ can be found by minimizing a single objective that attains zero upon convergence. This enables automatic model validation and hyperparameter tuning on the held-out set.

To sum up, the novel non-adversarial data alignment method presented in this paper combines the clear convergence criteria found in non-parametric and simple parametric approaches and the power of deep neural discriminators used in adversarial models. Our method finds a transformation of one dataset that makes it "equivalent" to another dataset with respect to the specified family of density models. We show that if that transformation is restricted to a normalizing flow, the resulting problem can be solved by minimizing a single simple objective that attains zero only if two domains are correctly aligned. We experimentally verify this claim and show that the proposed method preserves the local structure of the transformed distribution and that it is robust to model misspecification by both over- and under-parameterization. We show that minimizing the proposed objective is equivalent to training a particular adversarial network, but in contrast with adversarial methods, the performance of our model can be inferred from the objective value alone. We also characterize the the vanishing of generator gradient mode that our model shares with its adversarial counterparts, and principal ways of detecting it.

## 2 Log-Likelihood Ratio Minimizing Flow

In this section, we formally define the proposed method for aligning distributions. We assume that $M(\theta)$ is a family of densities parameterized by a real-valued vector $\theta$, and we fit models from $M$ to data by maximizing its likelihood across models from $M$. Intuitively, if we fit two models $\theta_A$ and $\theta_B$ to datasets $A$ and $B$ independently, and also fit a single shared model $\theta_S$ to the combined dataset $A \cup B$, then the log-likelihood ratio distance would equal the difference between the log-likelihood of that optimal "shared" and the two optimal "private" models (Definition 2.1). Next, we consider the problem of finding a transformation that would minimize this distance. In general, this would require solving an adversarial optimization problem (1), but we show that if the transformation is restricted to the family of normalizing flows, then the optimal one can be found by minimizing a simple non-adversarial objective (Theorem 2.3). We also illustrate this result with an example that can be solved analytically: we show that minimizing the proposed distance between two random variables

with respect to the normal density family is equivalent to directly matching their first two moments (Example 2.1). Finally, we show the relation between the proposed objective and Jensen-Shannon divergence and show that minimizing the proposed objective is equivalent to training a generative adversarial network with a particular choice of the discriminator family.

**Notation.** Let $\log P_M(X;\theta) := \mathbb{E}_{x\sim P_X}\log P_M(x;\theta)$ denote the negative cross-entropy between the distribution $P_X$ of the dataset $X$ defined over $\mathcal{X}\subset\mathbb{R}^n$, and a member $P_M(x;\theta)$ of the parametric family of distributions $M(\theta)$ defined over the same domain, i.e. the likelihood of $X$ given $P_M(x;\theta)$.

**Definition 2.1 (LRD).** *Let us define the log-likelihood ratio distance $d_\Lambda$ between datasets $A$ and $B$ from $\mathcal{X}$ with respect to the family of densities $M$, as the difference between log-likelihoods of $A$ and $B$ given optimal models with "private" parameters $\theta_A$ and $\theta_B$, and "shared" parameters $\theta_S$:*

$$d_\Lambda(A,B;M) = \max_{\theta_A;\theta_B}\Big[\log P_M(A;\theta_A)+\log P_M(B;\theta_B)\Big] - \max_{\theta_S}\Big[\log P_M(A;\theta_S)+\log P_M(B;\theta_S)\Big]$$

$$= \min_{\theta_S}\max_{\theta_A;\theta_B}\Big[\big(\log P_M(A;\theta_A)-\log P_M(A;\theta_S)\big)+\big(\log P_M(B;\theta_B)-\log P_M(B;\theta_S)\big)\Big].$$

The expression above is also the log-likelihood ratio test statistic $\log\Lambda_n$ for the null hypothesis $H_0:\theta_A=\theta_B$ for the model described by the likelihood function $P(A,B\mid\theta_A,\theta_B)=\big[P_M(A;\theta_A)\cdot P_M(B;\theta_B)\big]$ and intuitively equals to the amount of likelihood we "lose" by forcing $\theta_A=\theta_B$ onto the model fitted to approximate $A$ and $B$ independently. Figure 1 illustrates that, in terms of average likelihood, the shared model (purple) is always inferior to two private models from the same class, unless two datasets are in fact just different samples from the same distribution.

**Lemma 2.1.** *The log-likelihood ratio distance is non-negative, and the equals zero only if there exists a single "shared" model that approximates datasets as well as their "private" optimal models:*

$$d_\Lambda(A,B;M)=0 \Leftrightarrow \exists\,\theta_S:\log P_M(A;\theta_S)=\max_\theta\log P_M(A;\theta)\wedge\log P(B;\theta_S)=\max_\theta\log P_M(B;\theta).$$

*Proof.* Follows from the fact that the shared part in the Definition 2.1 is identical to the private part but over a smaller feasibility set $\{\theta_A=\theta_B\}$. See the supplementary Section 8.6 for the formal proof.

**Adversarial formulation.** If we introduce the parametric family of transformations $T(x,\phi)$ and try to find $\phi$ that minimizes the log-likelihood ratio distance $\min_\phi d_\Lambda(T(A;\phi),B;M)$, an adversarial problem arises. Note that for a fixed dataset $B$, only the first term is adversarial, and only w.r.t. $\theta_{AT}$:

$$\min_{\phi,\theta_S}\max_{\theta_{AT};\theta_B}\Big[\log P_M(T(A;\phi);\theta_{AT})+\log P_M(B;\theta_B)-\log P_M(T(A;\phi);\theta_S)-\log P_M(B;\theta_S)\Big]\quad(1)$$

Figure 1b illustrates that minimizing this objective (1) over $\theta_S$ while maximizing it over $\theta_{AT}$ corresponds to minimizing entropy ("squeezing") of the combination of $T(A)$ and $B$ while maximizing entropy of ("expanding") transformed dataset $T(A)$ as much as possible.

**Non-adversarial formulation.** The adversarial objective (1) requires finding a new optimal model $\theta_{AT}$ for each new value of $\phi$ to find the maximal likelihood of the transformed dataset $T(A)$, but Figure 1a illustrates that the likelihood of the transformed dataset can be often estimated from the parameters of the transformation $T$ alone. For example, if $T$ uniformly squeezes the dataset by a factor of two, the average maximum likelihood of the transformed dataset $\max_\theta\log P_M(T(A);\theta)$ doubles compared to the likelihood of the original $A$. In general, the likelihood of the transformed dataset is inversely proportional to the Jacobian of the determinant of the applied transformation. The lemma presented below formalizes this relation taking into account the limited capacity of $M$, and leads us to our main contribution: the optimal transformation can be found by simply minimizing a modified version of the objective (1) using an iterative method of one's choice.

**Lemma 2.2.** *If $T(x;\phi)$ is a normalizing flow, then the first term in the objective (1) can be bounded in closed form as a function of $\phi$ up to an approximation error $\mathcal{E}_{bias}$. The equality in (2) holds when the approximation term vanishes, i.e. if $M$ approximates both $A$ and $T(A;\phi)$ equally well; $P_A$ is the true distribution of $A$ and $T[P_A,\phi]$ is the push-forward distribution of the transformed dataset.*

$$\max_{\theta_{AT}}\log P_M(T(A;\phi);\theta_{AT}) \leq \max_{\theta_A}\log P_M(A;\theta_A)-\log\det|\nabla_x T(A;\phi)|+\mathcal{E}_{bias}(A,T,M)\quad(2)$$

$$\mathcal{E}_{bias}(A,T,M)\triangleq\max_\phi\Big[\min_\theta\mathcal{D}_{KL}(P_A;M(\theta))-\min_\theta\mathcal{D}_{KL}(T[P_A,\phi];M(\theta))\Big]$$

*Proof.* We expand likelihoods of combined and shared datasets given best models from $M$ into respective "true" negative entropies and the approximation errors due to the choice of $M$ (KL-divergence between true distributions and their KL-projections onto $M$). Then we replace the entropy of the transformed dataset with the entropy of the original and the log-determinant of the Jacobian of the applied transformation, noting that $\log\det|\nabla_x T^{-1}(T(A,\phi),\phi)| = \log\det|\nabla_x T(A,\phi)|$. We refer readers to the Section 8.6 of the supplementary for the full proof. $\square$

By applying this lemma to the objective (1) and grouping together terms that do not depend on $\theta_S$ and $\phi$, we finally obtain the final objective.

**Definition 2.2 (LMRF).** *Let us define the log-likelihood ratio minimizing flow (LRMF) for a pair of datasets $A$ and $B$ on $\mathcal{X}$, the family of densities $M(\theta)$ on $\mathcal{X}$, and the parametric family of normalizing flows $T(x;\phi)$ from $\mathcal{X}$ onto itself, as the flow $T(x;\phi^*)$ that minimizes $\mathcal{L}_{LRMF}$ (3), where the constant $c(A,B)$ does not depend on $\theta_S$ and $\phi$, and can be precomputed in advance.*

$$\mathcal{L}_{LRMF}(A,B,\phi,\theta_S) = -\log\det|\nabla_x T(A;\phi)| - \log P_M(T(A;\phi);\theta_S) - \log P_M(B;\theta_S) + c(A,B), \quad (3)$$
$$c(A,B) = \max_{\theta_A}\log P_M(A;\theta_A) + \max_{\theta_B}\log P_M(B;\theta_B)$$

**Theorem 2.3.** *If $T(x,\phi)$ is a normalizing flow, then the adversarial log-likelihood ratio distance (1) between the transformed source and target datasets can be bounded via the non-adversarial LRMF objective (3), and therefore the parameters of the normalizing flow $\phi$ that make $T(A,\phi)$ and $B$ equivalent with respect to $M$ can be found by minimizing the LRMF objective (3) using gradient descent iterations with known convergence guarantees.*

$$0 \le d_\Lambda(T(A,\phi),B;M) \le \min_\theta L_{LRMF}(A,B,\phi,\theta) + \mathcal{E}_{bias}. \quad (4)$$

This theorem follows from the definition of $d_\Lambda$ and two lemmas provided above that show that the optimization over $\theta_{AT}$ can be (up to the error term) replaced by a closed-form expression for the likelihood of the transformed dataset if the transformation is a normalizing flow. Intuitively, the LRMF loss (3) encourages the transformation $T$ to draw all points from $A$ towards the mode of the shared model $P(x,\theta_S)$ via the second term, while simultaneously encouraging $T$ to expand as much as possible via the first term as illustrated in Figure 1b. The delicate balance is attained only when two distributions are aligned, as shown in Lemma 2.1. The inequality (4) is tight (equality holds) only when the bias term is zero, and the shared model is optimal.

The example below shows that the affine log-likelihood ratio minimizing flow between two univariate random variables with respect to the normal density family $M$ corresponds to shifting and scaling one variable to match two first moments of the other, which agrees with our intuitive understanding of what it means to make two distributions "indistinguishable" for the Gaussian family.

---

**Example 2.1.** *Let us consider two univariate normal random variables $A, B$ with moments $\mu_A, \mu_B, \sigma_A^2, \sigma_B^2$, restrict $M$ to normal densities, and the transform $T(x;\phi)$ to the affine family: $T(x;a,b) = ax + b$, i.e. $\theta = (\mu,\sigma)$ and $\phi = (a,b)$. Using the expression for the maximum log-likelihood (negative entropy) of the normal distribution, and the expression for variance of the equal mixture, we can solve the optimization over $\theta_S = (\mu_S, \sigma_S)$ analytically:*

$$\min_{\mu,\sigma}\mathbb{E}_X\log P(X;\mu,\sigma) = -\frac{1}{2}\log(2\pi e\sigma_X^2) = -\log\sigma_X + C$$

$$\min_{\theta_S}\left[-\log P_M(T(A;\phi);\theta_S) - \log P_M(B;\theta_S)\right] = \log\left(\frac{1}{2}(a^2\sigma_A^2 + \sigma_B^2) - \frac{1}{4}(\mu_A + b - \mu_B)^2\right) - 2C.$$

*Combining expressions above gives us the final objective that can be solved analytically by setting the derivatives with respect to $a$ and $b$ to zero:*

$$\log\det|\nabla_x T(A;\phi)| = \log a \quad \text{and} \quad c(A,B) = -\log\sigma_A - \log\sigma_B + 2C,$$

$$\mathcal{L}_{LRMF} = -\log a + \log\left(\frac{1}{2}(a^2\sigma_A^2 + \sigma_B^2) - \frac{1}{4}(\mu_A + b - \mu_B)^2\right) - \log\sigma_A - \log\sigma_B$$

$$a^* = \frac{\sigma_B}{\sigma_A}, \quad b^* = \mu_B - \mu_A.$$

*The error term $\mathcal{E}_{bias}$ equals zero because any affine transformation of a Gaussian is still a Gaussian.*

---

**Relation to Jensen-Shannon divergence and GANs.** From the same expansion as in the proof of Lemma 2.2 and the information-theoretic definition of the Jensen-Shannon divergence (JSD) as the difference between entropies of individual distributions and their equal mixture, it follows that the likelihood-ratio distance (and consequently LRMF) can be viewed as biased estimates of JSD.

$$d_\Lambda(A, B) = 2 \cdot \mathrm{JSD}(A, B) - \mathcal{D}_{KL}(A, M) - \mathcal{D}_{KL}(B, M) + 2 \cdot \mathcal{D}_{KL}((A+B)/2, M)$$

Also, if the density family $M$ is "convex", in a sense that for any two densities from $M$ their equal mixture also lies in $M$, then by rearranging the terms in the definition of the likelihood-ratio distance, and noticing that the optimal shared model is the equal mixture of two densities, it becomes evident that the LRMF objective is equivalent to the GAN objective with the appropriate choice of the discriminator family:

$$\min_T d_\Lambda(T(A), B, M) = \min_T \max_{\theta_{AT}, \theta_B} \min_{\theta_S} \left[ \log \frac{P_M(T(A); \theta_{AT})}{P_M(T(A); \theta_S)} + \log \frac{P_M(B; \theta_B)}{P_M(B; \theta_S)} \right]$$

$$= \min_T \max_{\theta_{AT}, \theta_B} \left[ \log \frac{P_M(T(A); \theta_{AT})}{P_M(T(A); \theta_{AT}) + P_M(T(A); \theta_B)} + \log \frac{P_M(B; \theta_B)}{P_M(B; \theta_{AT}) + P_M(B; \theta_B)} + \log 4 \right]$$

$$= \min_T \max_{D \in \mathcal{H}} \left[ \log D(T(A)) + \log(1 - D(B)) + \log 4 \right], \quad \mathcal{H}(\theta, \theta') = \left\{ \frac{P_M(x; \theta)}{P_M(x; \theta) + P_M(x; \theta')} \right\}.$$

Since $M$ is not "convex" in most cases, minimizing the LRMF objective is equivalent to adversarially aligning two datasets against a regularized discriminator. From the adversarial network perspective, the reason why $\mathcal{L}_{\text{LRMF}}$ manages to solve this min-max problem using plain minimization is because for any flow transformation parameter $\phi$ the optimal discriminator between $T(A; \phi)$ and $B$ is defined in closed form: $D^*(x, \phi) = P_M(x; \theta_B^*) / \left( P_M(x; \theta_B^*) + P_M(T^{-1}(x; \phi); \theta_A^*) \det |\nabla_x T^{-1}(x; \phi)| \right)$.

**Vanishing of generator gradients.** The relation presented above suggests that the analysis performed by Arjovsky and Bottou [1] for GANs (Theorem 2.4, page 6) applies to LRMF as well, meaning that gradients of the LRMF objective w.r.t. the learned transformation parameters might vanish in higher dimensions. This implies that while the inequality (4) always holds, the model produces a useful alignment only when a sufficiently "deep" minimum of the LRMF loss (3) is found, otherwise the method fails, and the loss value should be indicative of this. An example presented below shows that reaching this deep minimum becomes exponentially more difficult as the initial distance between distributions grows, which is often the case in higher dimensions.

> **Example 2.2.** *Consider $M(\theta)$ that parameterizes all equal mixtures of two univariate Gaussians with equal variances, i.e. $\theta = (\mu_s^{(1)}, \mu_s^{(2)}, \sigma_s^2)$ and $P_M(x \mid \theta) = \frac{1}{2}\left( \mathcal{N}(x|\mu_s^{(1)}, \sigma_s^2) + \mathcal{N}(x|\mu_s^{(2)}, \sigma_s^2) \right)$. Consider $A$ sampled from $M(\delta, -\delta, \sigma_0^2)$ and $B_\mu$ sampled from $M(\mu + \delta, \mu - \delta, \sigma_0^2)$ for some fixed $\delta, \mu$ and $\sigma_0$. Let transformations be restricted to shifts $T(x; b) = x + b$, so $\phi = b$, and $\log \det |\nabla_x T(x; \phi)| = 0$, and $\mathcal{E}_{bias} = 0$ since $M$ can approximate both $A$ and $T(A; b)$ perfectly for any $b$. For a sufficiently large $\mu$, optimal shared model parameters can be found in closed form: $\theta^* = (b, \mu, \sigma_0^2 + \delta^2)$. This way the LRMF loss can be computed in closed form up to the cross-entropy: $L(b, \mu) := \min_\theta \mathcal{L}_{LRMF}(A, B_\mu, b, \theta) = -2H[M(\mu + \delta, \mu - \delta, \sigma_0^2), M(b, \mu, \sigma_0^2 + \delta^2)] + C$. A simulation provided in the supplementary Section 8.8 shows that the norm of the gradient of the LRMF objective decays exponentially as a function of $\mu$: $\|[\partial L(b, \mu)/\partial \mu](0, \mu)\| \propto \exp(-\mu^2)$, meaning that as $A$ and $B_\mu$ become further, the objective quickly becomes flat w.r.t $\phi$ near the initial $\phi_{t=0} = 0$.*

**Model complexity.** We propose the following intuition: 1) chose the family $M(\theta)$ that gives highest validation likelihood on $B$, since at optimum the shared model has to approximate the true underlying $P_B$ well; 2) chose the family $T(x; \phi)$ that has fewer degrees of freedom then $M$, since otherwise the problem becomes underspecified. For example, consider $M$ containing all univariate Gaussians parameterized by two parameters $(\mu, \sigma)$ aligned using polynomial transformations of the form $T(x; a_0, a_1, a_2) = a_2 x^2 + a_1 x + a_0$. In Example 2.1 we showed that Gaussian LRMF is equivalent to moment matching for two first moments, but with this choice of $T$, there exist infinitely many solutions for $\phi$ that all produce the desired mean and variance of the transformed dataset.

# 3 Related work

In this section we summarize the prior work on addressing domain adaptation as distribution alignment, recent advances in modeling probability densities using normalizing flows, and prior attempts at applying flows to domain adaptation and distribution discrepancy estimation.

**Domain Adaptation.** Ben-David et al. [4] showed that the test error of the learning algorithm trained and tested on samples from different distributions labeled using a shared "ground truth" labeling function is bounded by the $\mathcal{H}\Delta\mathcal{H}$-distance between the two distributions, therefore framing domain adaptation as distribution alignment. This particular distance is difficult to estimate in practice, so early neural feature-level domain adaptation methods such as deep domain confusion [26], DAN [15] or JAN [16] directly optimized estimates of non-parametric statistical distances (e.g. maximum mean discrepancy) between deep features of data points from two domains. Other early neural DA methods approximated domain distributions via simple parametric models, for example DeepCORAL [24] minimizes KL-divergence between pairs of Gaussians. Unfortunately, these approaches struggle to capture the internal structure of real-world datasets. Adversarial (GAN-based) approaches, such as ADDA [10] and DANN [27], address these limitations using deep convolutional domain discriminators. However, adversarial models are notoriously hard to train and provide few automated domain-agnostic convergence validation and model selection protocols, unless ground truth labels are available. Many recent improvements in the performance of classifiers adapted using adversarial alignment rely techniques utilizing source labels, such as semantic consistency loss [13], classifier discrepancy loss [22], or pseudo-labeling [9], added on top of the unsupervised adversarial alignment. The comparison to methods that use source labels is beyond the scope of this work, since we are primarily interested in improving the robustness of the underlying alignment method.

**Normalizing Flows.** The main assumption behind normalizing flows [20] is that the observed data can be modeled as a simple distribution transformed by an unknown invertible transformation. Then the density at a given point can be estimated using the change of variable formula by estimating the determinant of the Jacobian of that transformation at the given point. The main challenge in developing such models is to define a class of transformations that are invertible, rich enough to model real-world distributions, and simple enough to enable direct estimation of the aforementioned Jacobian determinant. Most notable examples of recently proposed normalizing flows include Real NVP [8], GLOW [14] built upon Real NVP with more general learnable permutations and trained at multiple scales to handle high resolution images, and the recent FFJORD [11], that used forward simulation of an ODE with an velocity field parameterized by a neural network as a flow transformation.

**Composition of inverted flows.** AlignFlow [12] is built of two flow models $G$ and $F$ trained on datasets $A$ and $B$ in the "back-to-back" composition $F \circ G^{-1}$ to map points from $A$ to $B$. We argue that the structure of the dataset manifold is destroyed if two flow are trained independently, since two independently learned "foldings" of lower-dimensional surfaces into the interior of a Gaussian ball are almost surely "incompatible" and render correspondences between $F^{-1}(B)$ and $G^{-1}(A)$ meaningless. Grover et al. [12] suggests to share some weights between $F$ and $G$, but we propose that this solution does not addresses the core of the issue. Yang et al. [29] showed that PointFlow - a variational FFJORD trained on point clouds of mesh surfaces - can be used to align these point clouds in the $F \circ G^{-1}$ fashion. But the point correspondences found by the PointFlow are again due to the spatial co-occurrence of respective parts of meshes (left bottom leg is always at the bottom left) and do not respect the structure of respective surface manifolds. Our approach requires 2-3 times more parameters then our composition-based baselines, but in the next section we show that it preserves the local structure of aligned domains better, and the higher number of trainable parameters does not cause overfitting.

**CycleGAN with normalizing flows.** RevGAN [28] used GLOW [14] to enforce the cycle consistency of the CycleGAN, and left the loss and the adversarial training procedure unchanged. We believe that the normalizing flow model for dataset alignment should be trained via maximum likelihood since the ability to fit rich models with plain minimization and validate their performance on held out sets are the primary selling points of normalizing flows that should not be dismissed.

**Likelihood ratio testing for out-of-distribution detection.** Nalisnick et al. [18] recently observed that the average likelihood is not sufficient for determining whether the given dataset came from the same distribution as the dataset used for training the density model. A recent paper by Ren et al. [19] suggested to use log-likelihood ratio test on LSTMs to *detect* distribution discrepancy in

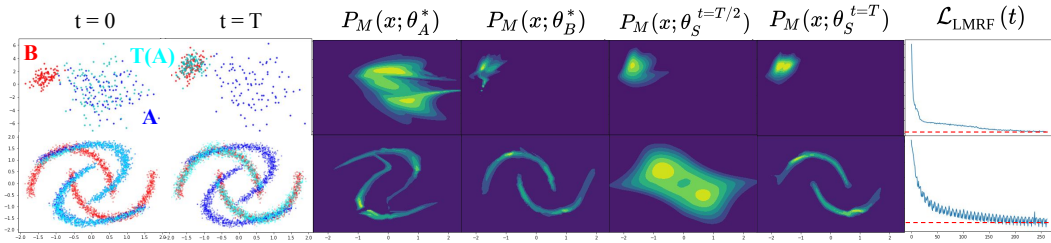

**Figure 2: The dynamics of training a Real NVP LRMF on the blob (first row) and moons (second row) datasets**. Blue, red and cyan points represent $A$, $B$ and $T(A)$ respectively. First two columns show $T(A)$ before and after training. Third and forth columns shows optimal models from $M$ for $A$ and $B$. Fifth and sixth columns show the evolution of the shared model. The last column shows the LRMF loss over time. Even a severely *overparameterized* LRMF does a good job at aligning blob distributions. The animated version that shows the evolution of respective models is available on the project web-page ai.bu.edu/lrmf. *Best viewed in color.*

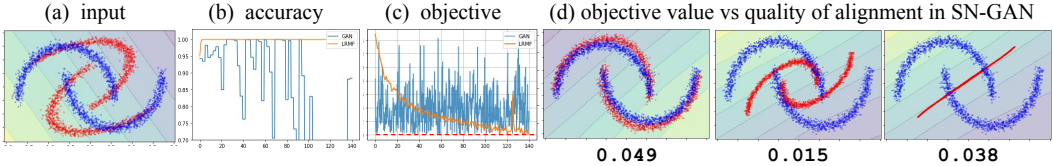

**Figure 3: The dynamics of training a GAN with Spectral Normalization (SN-GAN) on the moons dataset**. The adversarial framework provides means for aligning distributions against rich families of parametric discriminators, but requires the right choice of learning rate and an external early stopping criterion, because the absolute value of the adversarial objective (blue) is not indicative of the actual alignment quality even in low dimensions. The proposed LRMF method (orange) can be solved by plain minimization and converges to zero.

genomic sequences, whereas we propose a *non-adversarial* procedure for *minimizing* this measure of discrepancy using unique properties of normalizing flows.

## 4   Experiments and Results

In this section, we present experiments that verify that minimizing the proposed LRMF objective (3) with Gaussian, RealNVP, and FFJORD density estimators does indeed result in dataset alignment. We also show that both under- and over-parameterized LRMFs performed well in practice, and that resulting flows preserved the local structure of aligned datasets better then non-parametric objectives and the AlignFlow-inspired [12] baseline, and were overall more stable then parametric adversarial objectives. We also show that the RealNVP LRMF produced a semantically meaningful alignment in the embedding space of an autoencoder trained simultaneously on two digit domains (MNIST and USPS) and preserved the manifold structure of one mesh surface distribution mapped to the surface distribution of a different mesh. We provide Jupyter notebooks with code in JAX [6] and TensorFlow Probability (TFP) [7].

**Setup 1: Moons and blobs**. We used LRMF with Gaussian, Real NVP, and FFJORD densities $P_M(x; \theta)$ with affine, NVP, and FFJORD transformations $T(x; \phi)$ respectively to align pairs of moon-shaped and blob-shaped datasets. The blobs dataset pair contains two samples of size $N = 100$ from two Gaussians. The moons dataset contains two pairs of moons rotated $50°$ relative to one another. We used original hyperparameters and network architectures from Real NVP [8] and FFJORD [11], the exact values are given in the supplementary. We also measured how well the learned LRMF transformation preserved the local structure of the input compared to other common minimization objectives (EMD, MMD) and the "back-to-back" composition of flows using a 1-nearest neighbor classifier trained on the target and evaluated on the transformed source. We also compared our objective to the adversarial network with spectral normalized discriminator (SN-GAN) in terms of how well their alignment quality can be judged based on the objective value alone.

**Results.** In agreement with Example 2.1, affine Gaussian LRMF matched first two moments of aligned distributions (Figure 8). In Real NVP (Figures 2,9) and FFJORD (Figure 10) experiments the shared model converged to $\theta_B^*$ gradually "enveloping" both domains and pushing them towards each other. In both under-parameterized (Gaussian LRMF on moons) and over-parameterized (RealNVP LRMF on blobs) regime our loss successfully aligns distributions. In all experiments, the LRMF loss converged to zero in average (red line), so $\mathcal{E}(A, T, M) \approx 0$, meaning that affine, Real NVP, and FFJORD transformations keep input distributions "equally far" from $M$. The loss occasionally dropped below zero because of the variance in mini-batches. Figure 4 shows that, despite good

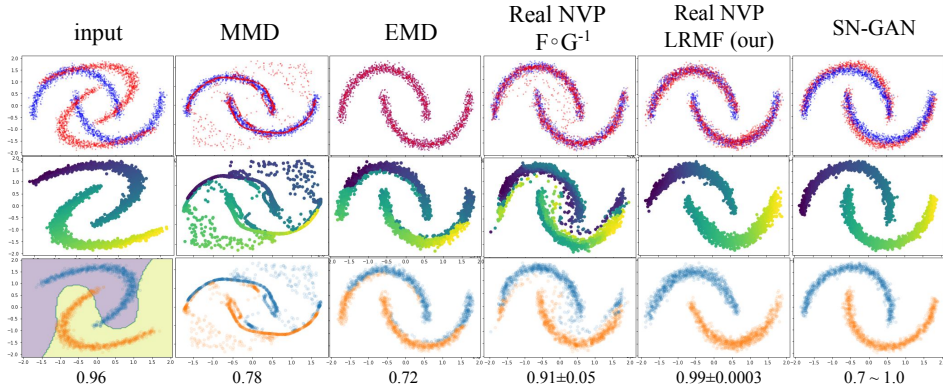

| input | MMD | EMD | Real NVP F∘G⁻¹ | Real NVP LRMF (our) | SN-GAN |
|---|---|---|---|---|---|
| 0.96 | 0.78 | 0.72 | 0.91±0.05 | 0.99±0.0003 | 0.7 ~ 1.0 |

**Figure 4: Among the non-adversarial alignment objectives, only LRMF preserves the manifold structure of the transformed dataset**. Each domain contains two moons. The top row shows how well two domains (red and blue) are aligned by different methods trained to transform the red dataset to match the blue dataset. The middle row shows new positions of points colored consistently with the first column. The bottom row shows what happens to red moons after the alignment. Numbers at the bottom of each figure show the accuracy of the 1-nearest neighbor classifier trained on labels from the blue domain and evaluated on transformed samples from the red domain. The animated version is available on the project web-page http://ai.bu.edu/lrmf.

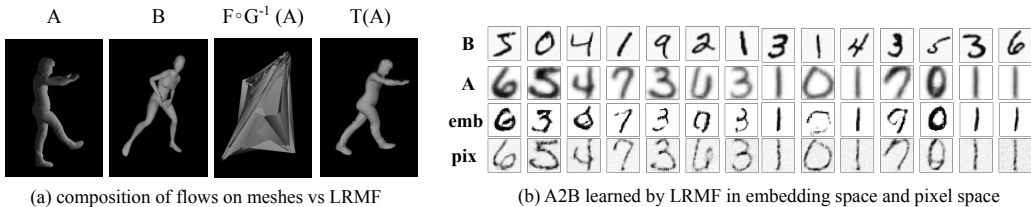

(a) composition of flows on meshes vs LRMF      (b) A2B learned by LRMF in embedding space and pixel space

**Figure 5: RealNVP LRMF successfully semantically aligned digits and preserved the local structure of the mesh surface manifold**. **(a)** The marginal distribution produced by the "back-to-back" composition $F \circ G^{-1}$ of two normalizing flows trained on vertices of two meshes matches the point distribution of $B$, but the local structure of the original manifold is distorted, while LRMF preserves the local structure. **(b)** USPS digits ($B$) transformed into MNIST digits ($A$) via LRMF in VAE embedding space (*emb*), via LRMF in pixel space (*pix*).

marginal alignment (top row) produced by MMD, EMD, and the $F \circ G^{-1}$ composition (inspired by AlignFlow [12]), the alignment produced by LRMF preserved the local structure of transformed distributions better, comparably to the SN-GAN both qualitatively (color gradients remain smooth in the middle row) and quantitatively in terms of adapted 1-NN classifier accuracy (bottom row). We believe that LRMF and SN-GAN preserved the local structure of presented datasets better than non-parametric models because assumptions about aligned distributions are too general in the non-parametric setting (overall smoothness, etc.), i.e. parametric models (flows, GANs) are better at capturing structured datasets. At the same time, Figure 3 shows that the quality of the LRMF alignment can be judged from the objective value (orange line) and stays at optima upon reaching it, while SN-GAN's performance (blue) can be hardly judged from the value of its adversarial objective and diverges even from near-optimal configurations.

**Setup 2: Meshes.** We treated vertices from two meshes as samples from two mesh surface point distributions and aligned them. After that, we draw faces of the original mesh at new vertex positions. We trained two different flows $F$ and $G$ on these surface distributions, and passed one vertex cloud through their back-to-back composition, and compared this with the result obtained using LRMF.

**Results.** Figure 5a shows that, in agreement with the previous experiment, the number of points in each sub-volume of $B$ matches the corresponding number in the transformed point cloud $F(G^{-1}(A))$, but drawing mesh faces reveals that the local structure of the original mesh surface manifold is distorted beyond recognition. The LRMF alignment (fourth column) better preserves the local structure of the original distribution - it rotated and stretched $A$ to align the most dense regions (legs, torso, head) with the most dense regions of $B$.

**Setup 3: Digit embeddings.** We trained a VAE-GAN to embed unlabeled images from USPS and MNIST into a shared 32-dimensional latent space. We trained a Real NVP LRMF to map latent codes of USPS digits to latent codes of MNIST. We also trained digit label classifiers on images obtained by decoding embeddings transformed using LRMF, CORAL, and EMD and applied the McNemar test of homogeneity [17] to the contingency tables of prediction made by these classifiers.

**Results.** The LRMF loss attained zero. Figure 5b(emb) shows that LRMF semantically aligned images form two domains. Classifiers trained on images transformed using LMRF had higher

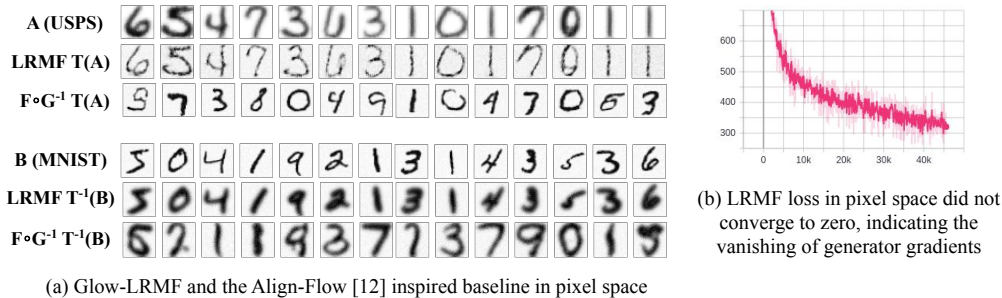

(a) Glow-LRMF and the Align-Flow [12] inspired baseline in pixel space

(b) LRMF loss in pixel space did not converge to zero, indicating the vanishing of generator gradients

**Figure 6: GLOW-LRMF did not converge in pixel space, but preserved class labels much better then the AlignFlow-inspired baseline [12].** **(a)** Images generated by applying learned flow models in forward and backward direction to USPS and MNIST respectively. **(b)** GLOW-LRMF loss did not converge to zero due to the vanishing gradients in higher dimensions (pixel space). This failure mode can be detected by looking at the loss values alone.

accuracy on the target dataset (.55 for LRMF vs .47 for EMD vs .48 for CORAL). McNemar test showed that LRMF's improvements in accuracy were significant (p-value $\ll$ 1e-3 in all cases).

**Setup 4: GLOW.** We trained a GLOW LRMF to align USPS and MNIST in 32x32 pixel space, visualized outputs of the forward and backward transformation, and the LRMF loss value over training iterations.

**Results.** The model learned to match the stroke width across domains, but did not make images completely indistinguishable (Figure 6). The shared density model converged to the local minima that corresponds to approximating $T(A)$ and $B$ as two distinct "bubbles" of density that fail to merge. This is the same failure mode we illustrated in Example 2.2 where two components of the shared model get stuck approximating datasets that are too far away, and fail to bring the model into the deeper global minima. We would like to note that even though the loss did not converge to zero, i.e. the model failed to find a marginally perfect alignment, it did so *not silently*, in stark contrast with adversarial methods that typically fail silently. These results agree with our hypothesis about vanishing transformation gradients in higher dimensions (end of Section 2), resulting in vast flat regions in the LRMF loss landscape with respect to the transformation parameter $\phi$, obstructing full marginal alignment. The AligFlow-inspired [12] composition of flows ($F \circ G^{-1}$ in Figure 6), on the other hand, produced very good marginal alignment, judging from the fact that transformed images look very much like MNIST and USPS digits, but erased the semantics in the process, judging from the mismatch between classes of original and transformed digits.

## 5 Conclusion and Future Work

In this paper, we propose a new alignment objective parameterized by a deep density model and a normalizing flow that, when converges to zero, guarantees that the density model fitted to the transformed source dataset is optimal for the target and vice versa. We also show that the resulting model is robust to model misspecification and preserves the local structure better than other non-adversarial objectives. We showed that minimizing the proposed objective is equivalent to training a particular GAN, but is not subject to mode collapse and instability of adversarial training, however in higher dimensions, is still affected by the vanishing of generator gradients. Translating recent advances in dealing with the vanishing of generator gradients, such as instance noise regularization [1; 21; 23], to the language of likelihood-ratio minimizing flows offers an interesting challenge for future research.

## 6 Acknowledgements

This work was partially supported by NSF award #1724237, DARPA and Google.

## 7 Broader Impact

Many recent advances in deep learning rely heavily on large labeled datasets. Unfortunately, in many important problem domains, such as medical imaging, labeling costs and high variability of target environments, such as differences in image capturing medical equipment, prohibit widespread adoption of these novel deep image processing techniques.

Our work proposes a deep domain adaptation method that brings together verifiable convergence, as in older non-parametric methods, and meaningful priors over the structure of aligned datasets from deep adversarial alignment models.

Of course, none of aforementioned advancements can guarantee perfect semantic alignment, therefore manual evaluation in critical applications, such as medical diagnosis, is still required. However, improved interpretability that comes from having a single minimization objective would definitely ease the adoption of such methods by making validation and model selection more straightforward, as well as reducing the chance of deploying a silently failing model due to human evaluator error.

As with the majority of deep models, our model might be susceptible to adversarial attacks by malicious agents, as well as privacy-related attacks, but properly addressing these issues, their consequences, and defence techniques goes far beyond the scope of this paper.

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
