[Supplementary Material]

# 8 Supplementary Material

## 8.1 Pseudo-code for the learning algorithm

---

**Algorithm 1:** Mini-batch training of Log-Likelihood Ratio Minimizing Flow

---

**inputs:** datasets $A$ and $B$; normalizing flow model $T(x; \phi)$; density model $P_M(x; \phi)$;

  learning rate $\eta$; thresholds $(\epsilon, \varepsilon)$; batch size $b$; initial parameters values $\phi^{(0)}, \theta_A^{(0)}, \theta_B^{(0)} \theta_{AT}^{(0)}, \theta_S^{(0)}$;

**outputs:** convergence indicator $I_c$; weights $\phi^*$ that make $T(A; \phi^*)$ and $B$ equivalent w.r.t. M;

**foreach** $X \in \{A, B\}$ **do**

    $t \leftarrow 0$ ;                              `// first learn optimal models` $\theta_A^*, \theta_B^*$

    **while** $||\nabla_\theta \log P_M(X; \theta_X^{(t)})|| \geq \varepsilon$ **do**

        $x^{(t)} \leftarrow$ draw batch of size $b$ from $X$;

        $\theta_X^{(t+1)} \leftarrow \theta_X^{(t)} + \eta \cdot \nabla_\theta \log P_M(x^{(t)}; \theta_X^{(t)})$;

        $t \leftarrow t + 1$;

    **end while**

    $\theta_X^* \leftarrow \theta_X^{(t)}$

**end foreach**

$t \leftarrow 0$ ;                                        `// now train LRMF`

**while** $||g_T^{(t)}|| + ||g_S^{(t)}|| \geq \varepsilon$ **do**

    $A^{(t)} \leftarrow$ draw batch of size $b$ from $A$;

    $B^{(t)} \leftarrow$ draw batch of size $b$ from $B$;

    $g_S^{(t)} \leftarrow \nabla_\theta \left[ \log P_M(T(A^{(t)}; \phi^{(t)}); \theta_S^{(t)}) + \log P_M(B^{(t)}; \theta_S^{(t)}) \right]$;

    $g_T^{(t)} \leftarrow \nabla_\phi \left[ \log P_M(T(A^{(t)}; \phi^{(t)}); \theta_S^{(t)}) + \log \det |\nabla_x T(A^{(t)}; \phi^{(t)})| \right]$;

    $\theta_S^{(t+1)} \leftarrow \theta_S^{(t)} + \eta \cdot g_S^{(t)}$;

    $\phi^{(t+1)} \leftarrow \phi^{(t)} + \eta \cdot g_T^{(t)}$;

    $t \leftarrow t + 1$;

**end while**

$\theta_S^* \leftarrow \theta_S^{(t)}$;

$\phi^* \leftarrow \phi^{(t)}$;

$c_{AB} \leftarrow \log P_M(A; \theta_A^*) + \log P_M(B; \theta_B^*)$ ;            `// and check convergence`

$\mathcal{L}_{\text{LRMF}} \leftarrow c_{AB} - \log P_M(T(A; \phi^*); \theta_S^*) - \log P_M(B; \theta_S^*) - \log \det |\nabla_x T(A; \phi^*)|$;

**if** $\mathcal{L}_{LRMF} \geq \epsilon$ **then** $I_c \leftarrow$ failed;

**else** $I_c \leftarrow$ succeeded;

**return** $(I_c, \phi^*)$

---

## 8.2 Attached code

Attached IPython notebooks were tested to work as expected in Colab. The JAX version (`lrmf_jax_public.ipynb`) includes experiments on 1D and 2D Gaussians and Real NVP, the Tensorflow Probabiliy (TFP) version (`lrmf_tfp_public.ipynb`) includes experiments on Real NVP and FFJORD. Files `vae_gan_public.ipynb`, `lrmf.py` and `lrmf_glow_public.ipynb` contain code we used for VAE-GAN training and GLOW LRMF training

## 8.3 Hyper-parameters

**Data.** Blobs datasets were samples from 2-dimensional Gaussians with parameters

$$\mu_A = \begin{bmatrix} 1.0 \\ 1.0 \end{bmatrix}, \Sigma_A^{-\frac{1}{2}} = \begin{bmatrix} 0.5 & 0.7 \\ -0.5 & 0.3 \end{bmatrix}$$

$$\mu_B = \begin{bmatrix} 4.0 \\ -2.0 \end{bmatrix}, \Sigma_B^{-\frac{1}{2}} = \begin{bmatrix} 0.5 & 3.0 \\ 3.0 & -2.0 \end{bmatrix}.$$

The moons dataset contains two pairs of moons rotated $50°$ relative to one another generate via `sklearn.datasets.make_moons` with $\varepsilon = 0.05$ containing 2000 samples each.

**Model.** In the affine LRMF with Gaussian density experiment (Figure 8), we parameterized the positive-definite transformation as $T(x, A, b) = A^T A \cdot x + b$ and the Gaussian density with parameters $(\mu, \Sigma^{-\frac{1}{2}})$ to ensure that $\Sigma$ is always positive definite as well. In Real NVP experiments we stacked four NVP blocks (spaced by permutations), each block parameterized by a dense neural network for predicting shift and scale with two 512-neuron hidden layers with ReLUs (the "default" Real NVP). In VAE-GAN experiments we trained a VAE-GAN on In FFJORD experiments we stacked two FFJORD transforms parameterized by DNN with `[16, 16, 16, 2]` hidden layers with hyperbolic tangent non-linearities. For the GLOW experiment we stacked three GLOW transformations at different scales each with eight affine coupling blocks spaced by act norms and permutations each parameterized by a CNN with two hidden layers with 512 filters each. In the GLOW experiment we parameterized $T$ as the back-to-back composition of same flows used for density estimation, but initialized from scratch instead of optimal models for $A$ and $B$. We used the Adam optimizer with learning rate $10^{-5}$ for training.

## 8.4 Other design considerations

### On the relation to the Invariant Risk Minimization.

In a recent arXiv submission, Arjovsky et al. [2] suggested that in the presence of an observable variability in the environment $e$ (e.g. labeled night-vs-day variability in images) the representation function $\Phi(x)$ that minimizes the conventional empirical risk across all variations actually yields a subpar classifier. One interpretation of this statement is that instead of searching for a representation function $\Phi(x)$ that minimizes the expected value of the risk

$$\mathcal{R}^e(f) = \mathbb{E}_{(X,Y) \sim P_e} l(f(X), Y)$$

across all variations in the environment $e$:

$$\min_\Phi \min_\theta \mathbb{E}_e \mathcal{R}^e(f(\Phi(\cdot), \theta))$$

one should look for a representation that is optimal under each individual variation of the environment

$$\min_\Phi \left[ \min_\theta \mathbb{E}_\epsilon \mathcal{R}^e(f(\Phi(\cdot), \theta)) - \mathbb{E}_\epsilon \min_{\theta_e} \mathcal{R}^e(f(\Phi(\cdot), \theta_e)) \right]$$

Arjovsky et al. [2] linearise this objective combined with the conventional ERM around the optimal $\theta$, and express the aforementioned optimally across all environments as a gradient penalty term that equals zero only if $\Phi$ is indeed optimal across all environment variations:

$$\min_\Phi \min_{\theta'} \mathbb{E}_e \mathcal{R}^e(f(\Phi(\cdot), \theta')) + \lambda \mathbb{E}_e ||\nabla_\theta \mathcal{R}^e(f(\Phi(\cdot), \theta'))||_2.$$

If we perform the Taylor expansion of the log-likelihood ratio statistic near the optimal shared model $\theta_S$, we get the score test statistic - a "lighter" version of the log-likelihood ratio test that requires training only a single model. Intuitively, if we train a model from $M$ simultaneously on two datasets $A$ and $B$ until convergence, i.e. until the average gradient of the loss w.r.t. weights $g_X = \nabla_\theta L(X; \theta)$ summed across both datasets becomes small $||g_A + g_B|| \leq \varepsilon$, then the combined norm of two gradients computed across each dataset independently would be small $||g_A|| + ||g_B|| \leq \varepsilon$, only under the null hypothesis ($A$ and $B$ are equivalent w.r.t. $M$). From our experience, this approach works well for detecting the presence of the domain shift, but is hardly suitable for direct minimization.

Both procedures and resulting objectives are very much reminiscent of the log-likelihood ratio minimizing flow objective we propose in this paper, and we would have obtained the score test version

if we linearized our objective around the optimal $\theta_S$. The main difference being that Arjovsky et al. [2] applied the idea of invariance across changing environments to the setting of supervised training via risk minimization, whereas we apply it to unsupervised alignment via likelihood maximization.

**On directly estimating likelihood scores across domains.**

One could suggest to estimate the similarity between datasets by directly evaluating and optimizing some combination of $P_M(A; \theta_B)$ and $P_M(B; \theta_A)$. Unfortunately, high likelihood values themselves are not very indicative of belonging to the dataset used for training the model, especially in higher dimensions, as explored by Nalisnick et al. [18]. One intuitive example of this effect in action is that for a high-dimensional normally distributed $x \sim \mathcal{N}_d(0, I)$ the probability of observing a sample in the neighbourhood of zero $P(||x|| \leq r)$ is small, but if we had a dataset $\{y_i\}_{i=0}^n$ sampled from that neighbourhood $||y_i|| \leq r$, its log-likelihood $\sum_i \log \mathcal{N}_d(y_i|0, I)$ would be high, even higher then the likelihood of the dataset sampled from $\mathcal{N}_d(0, I)$ itself. The proposed method, however, is not susceptible to this issue as we always evaluate the likelihood on the same dataset we used for training.

**On matching the parameters of density models.**

Two major objections we have to directly minimizing the distance between parameters $\theta$ of density models fitted to respective datasets $||\theta_{AT} - \theta_B||$ are that: a) the set of parameters that describes a given distribution might be not unique, and this objective does not consider this case; and b) one would have to employ some higher-order derivatives of the likelihood function to account for the fact that not all parameters contribute equally to the learned density function, therefore rendering this objective computationally infeasible to optimize for even moderately complicated density models.

**On replacing the Gaussian prior with a learned density in normalizing flows.**

We explored whether a similar distribution alignment effect can be achieved by directly fitting a density model to the target distribution $B$ to obtain the optimal $\theta_B^*$ first, and then fitting a flow model $T(x, \phi)$ to the dataset $A$ but replacing the Gaussian prior with the learned density of $B$:

$$\max_\phi \Big[ \log P_M(T^{-1}(A, \phi); \theta_B^*) - \log \det |\nabla_x T(A; \phi)| \Big].$$

While this procedure worked on distributions that were very similar to begin with, in the majority of cases the log-likelihood fit to $B$ did not provide informative gradients when evaluated on the transformed dataset, as the KL-divergence between distributions with disjoint supports is infinite. Moreover, even when this objective did not explode, multi-modality of $P_M(x; \theta_B)$ often caused the learned transformation to map $A$ to one of its modes. Training both $\phi$ and $\theta_B$ jointly or in alternation yielded a procedure that was very sensitive to the choice of learning rates and hyperparameters, and failed silently, which were the reasons we abandoned adversarial methods in the first place. The LRMF method described in this paper is not susceptible to this problem, because we never train a density estimator on one dataset and evaluate its log-likelihood on another dataset.

### 8.5 FFJORD LRMF experiment on moons.

As mentioned in the main paper, FFJORD LRMF performed on par with Real NVP version. We had to fit $T(x, \phi)$ to identity function prior to optimizing the LRMF objective, because the glorot uniform initialized 5-layer neural network with tanh non-linearities (used as a velocity field in FFJORD) generated significantly non-zero outputs. The dynamics can be found in the Figure 10.

### 8.6 Proof of Lemma 2.1

*Proof.* If we define $f(x) = \log P_M(A, x)$ and $g(x) = \log P_M(B, x)$, the first statement $d_\Lambda \geq 0$ follows from the fact that

$$\forall x \; f(x) + g(x) \geq \min_x f(x) + \min_x g(x) \; \Rightarrow \; \min_x(f(x) + g(x)) - \min_x f(x) - \min_x g(x) \geq 0$$

The second statement $f(x^*) = \min_x f(x), g(x^*) = \min_x g(x)$ comes form the fact that the equality holds only if there exists such $x^*$ that

$$f(x^*) + g(x^*) = \min_x f(x) + \min_x g(x)$$

Assume that $f(x^*) \neq \min_x f(x)$, then $f(x^*) > \min_x f(x)$ from the definition of the $\min$, therefore

$$g(x^*) = (f(x^*) + g(x^*)) - f(x^*) < (\min_x f(x) + \min_x g(x)) - \min_x f(x) = \min_x g(x),$$

which contradicts the definition of the $\min_x g(x)$, therefore $f(x^*) = \min_x f(x)$. $\qquad\square$

**(a)** $\sqrt{-\log(|\partial L/\partial \mu|)}$ vs $a\mu + b$  **(b)** $|\partial L/\partial \mu|$ vs $\exp(-(a\mu+b)^2)$  **(c)** same as **b** for $\mu \in 7 \ldots 16$

**Figure 7:** Gradient of the cross-entropy of between two mixture models as a function of the mean of one of the first components of the first mixture to illustrate the Example 2.2, estimated using JAX.

## 8.7  Proof of Lemma 2.2

First, we add and remove the true (unknown) entropy $H[P_A] = -\mathbb{E}_{a \sim P_A} \log P_A(a)$:

$$\max_{\theta_A} \mathbb{E}_{a \sim P_A} \log P_M(a; \theta_A) = \max_{\theta_A} \left[ \mathbb{E}_{a \sim P_A} \log P_A(a) - \mathbb{E}_{a \sim P_A} \log \frac{P_A(a)}{P_M(a; \theta_A)} \right]$$

$$= H[P_A] - \min_{\theta_A} \mathbb{E}_{a \sim P_A} \left[ \log \frac{P_A(a)}{P_M(a; \theta_A)} \right] = H[P_A] - \min_{\theta} \mathcal{D}_{KL}(P_A; M(\theta)). \quad (\star)$$

And then add and remove the (unknown) entropy of the transformed distribution $H[T[P_A, \phi]]$. We also use the change of variable formula $T[P_A](x) = P_A(T^{-1}(x)) \cdot \det |\nabla_x T^{-1}(x)|$, and substitute the expression for $H[P_A]$ from the previous line $(\star)$:

$$\max_{\theta_{AT}} \log P_M(T(A; \phi); \theta_{AT}) = \max_{\theta_{AT}} \mathbb{E}_{a' \sim T[P_A, \phi]} \log P_M(a'; \theta_{AT})$$

$$= \max_{\theta_{AT}} \left[ \mathbb{E}_{a' \sim T[P_A, \phi]} \log T[P_A](a') - \mathbb{E}_{a' \sim T[P_A, \phi]} \log \frac{T[P_A, \phi](a')}{P_M(a'; \theta_{AT})} \right]$$

$$= \max_{\theta_{AT}} \Big[ \mathbb{E}_{a \sim P_A} P_A(T^{-1}(T(a, \phi), \phi)) +$$

$$+ \log \det |\nabla_x T^{-1}(T(a, \phi), \phi)| - \mathcal{D}_{KL}(T[P_A, \phi]; M(\theta_{AT})) \Big]$$

$$= H[P_A] - \log \det |\nabla_x T(A, \phi)| - \min_{\theta} \mathcal{D}_{KL}(T[P_A, \phi]; M(\theta))$$

$$\leq \max_{\theta_A} \log P_M(A; \theta_A) - \log \det |\nabla_x T(A, \phi)| + \mathcal{E}_{bias}(A, T, M).$$

## 8.8  Simulation results for the Example 2.2

We approximated $|\partial H[m_1, m_2(\mu)]/\partial \mu|$, where $m_1$ and $m_2(\mu)$ are two equal mixtures of normal distributions, by computing the partial derivative using auto-differentiation in JAX. The objective was $L = \text{logsumexp}(\{\log(p_i(X; \mu)) + \log 2\}_i)$, where $\log p_i(x; \mu)$ is a log probability of the mixture component from $m_2$, and $X$ is a fixed large enough (n=100k) sample from the $m_1$. Figure 7 shows that $\sqrt{-\log(|\partial L/\partial \mu|)}$ fits to $a\mu + b$ for $a = 0.6, b = -1.168$ with $R = 0.99996$, therefore making us believe that $\|[\partial L(b, \mu)/\partial \mu](0, \mu)\| \propto \exp(-\mu^2)$. The code is available in `lrmf_gradient_simulation.ipynb`.

**Figure 8: The dynamics of training an affine log-likelihood ratio minimizing flow (LRMF) w.r.t. the Gaussian family on the blob and moons datasets**. The LRMF is trained to match $A$ (blue) with $B$ (red), its outputs $T(A)$ are colored with cyan, circles indicate $3\sigma$ levels of $\theta_A, \theta_B$ and $\theta_{AT}$ respectively. This experiment shows that even a severely *under-parameterized LRMF* does a good job at aligning distributions (second row). As in the **Example 2.1**, the optimal affine LRMF w.r.t. Gaussian family matches first two moments of given datasets. Rightmost column shows LRMF convergence.

(i) blobs

(ii) moons

**Figure 9: The dynamics of training a Real NVP log-likelihood ratio minimizing flow (LRMF) on the blob and moons datasets**. This experiment shows that even a severely *overparameterized LRMF* does a good job at aligning distributions: RealNVP clearly overfits to the blob dataset but learns a good alignment nevertheless. **(a, b)** The Real NVP density estimators fitted to datasets A (blue) and B (red). **(c)** The LRMF objective (Eq 2) decreases over time and reaches zero when two datasets are aligned. The red line indicates the zero loss level. **(d)** The evolution $A$ (blue), $A' = T(A, \phi)$ (cyan) and $B$ (red). **(e)** The probability density function of the shared model $P_M(x, \theta_S)$ fitted to $A'$ and $B$. When LRMF objective converges, $P_M(x, \theta_S)$ matches $P_M(x, \theta_B)$. **(f)** The visualization of the trained normalizing flow $T$, at each point $x$ we draw a vector pointing along the direction $v = x - T(x, \phi)$ with color intensity and length proportional to $v$.

**Figure 10: The dynamics of training a FFJORD log-likelihood ratio minimizing flow (LRMF) on the moons dataset**. Notations are similar to Figures 5 and 6. The left bottom plot shows changes in accuracy over time.

(a) evolution of T(A) over time across batches

(b) the value of the LRMF objective

**Figure 11: The dynamics of RealNVP LRMF semantically aligning USPS and MNIST digits in the latent space**. (a) Different rows in $T(A_0, t)$ represent transformations of the batch $A_0$ different time steps. Other rows represent the final learned transformation applied to other batches $A_1, A_2$. (b) The LRMF objective converged to zero in average.