[Reviews · NeurIPS 2020]

Review 1

Summary and Contributions: This paper proposes a new method for distribution alignment which utilises a normalising flow as alignment transformation. The authors propose a new objective (LRMF) which bounds the log-likelihood ratio distance. The main advantage over existing objectives is that LRMF is non-adversarial. As a result, the authors claim that training is more robust and that the model has a meaningful loss metric that can be used to show convergence.

Strengths: The paper is well-written and utilizes theory very well to explain the method. The method itself is an important addition to literature, as it introduces a direct optimization method for distribution alignment, which is typically done by adversarial methods. The theory in the paper is a pleasure to read, because the motivation for different concepts is explained well. In addition, the paper illustrates problems and methods using examples and figures, which aids understanding. The paper places itself in context by comparing to existing methods in a structured manner, and discusses advantages and disadvantages. The paper is also upfront about limitations (mainly dimensionality leading to vanishing gradients).

Weaknesses: The main weakness of the paper is the experimental section. Whereas adversarial methods in literature manage to align complex input images, the paper mainly focuses on smaller problems. In Figure 5 a,b, it could be made clearer that T(A) and emb/pix refer to the method of the authors. Also it is unclear whether the F G^{-1}(A) composition in the experiments refers to an AlignFlow-inspired baseline, please clarify this. As the authors discuss, scalability to higher dimensions of this method is its main limitation in its current form. The proofs and claims in the paper seem to depend on being able to find global optima (for instance eq. 3 depends on finding the optimal parameters for theta). Since in practice neural networks do not really converge to global (or even local) optima, it is currently unclear how optimization effects these claims. Additionally, it would be helpful if the source of the second inequality in Eq. 3 (between d_Lambda and min L_LRMF) could be quantified. When does equality hold (apart from the trivial case when L_LRMF and Epsilon_bias are zero)?

Correctness: The claims in the paper seem to be correct. Empirical methodology seems to be correct.

Clarity: The paper is really well-written. I would recommend this paper to newcomers in the field because of the illustrated introduction and the clearly outlined related work section.

Relation to Prior Work: The relation the prior work is clearly discussed. The related work section is structured nicely into different existing approaches.

Reproducibility: Yes

Additional Feedback: Since the paper utilizes a flow as distribution transformation, it is not possible to change dimensionality. The authors could address this limitation and discuss possible work-arounds. --------- After rebuttal ------- I have read the rebuttal and I am satisfied with the answers of the authors. I will keep my score as it is.


Review 2

Summary and Contributions: The paper proposed a method for neural distribution alignment with likelihood ratio as cost function, and show that under certain transformation family, i.e., normalising flows, this cost can be optimised efficiently compared to existing methods of neural distribution alignment, and also provides an interpretable value for model selection. The intuition behind the approach is that two datasets are indistinguishable from each other in the context of a model family if the combined likelihood of the two dataset under individual fitted model is the same as the likelihood of the combined dataset under a single fitted model. Additionally, the likelihood of the fitted individual model for transformed dataset can be approximated in closed form under a suitable family of transformation, e.g., normalising flow where the Jacobian is well formed. Both these observations lead to the final objective function (Def 2.2) that involves maximisation over two parameters, i.e., parameters of the shared model and the parameters of the normalising flow respectively.

Strengths: The paper tackles a relevant problem and provides an intriguing perspective. The arguments are well supported by examples, explanations and illustrations. The connection between the LRMF and existing methods is very relevant.

Weaknesses: The experiments on real dataset is limited and can be extended further.

Correctness: The derivation of method seems to be correct, and experimental evidence is convincing although section on real data can be extended.

Clarity: The paper is very well written, and quite easy to follow with well placed examples and illustrations.

Relation to Prior Work: The paper addressed prior work extensively, and explicitly discuss the relationship between the proposed model and existing models.

Reproducibility: Yes

Additional Feedback: Should this be likelihood ratio than log-likelihood ratio given the authors are taking the log of likelihood ratio (definition 2.1) than ratio of log-likelihood? The description of experiments on real datasets is too short. It will be great to provide some more information and enlarge the figures. The authors discuss the experiment on simulated dataset in great detail, and provide thoughtful insight and comparison. However, similar level of details and comparison are missing on the real datasets, for example, why the experiment was selected, do LRMF perform better, why digits were aligned in both embedding and pixel space? Some more information on what transformation has been used will be great. For example, what is RealNVP etc., and how were specific transformations chosen in each experiment? Figure 2. The boundary between the first and second row is not obvious. ---------- I have read the rebuttal and I thank the authors for addressing my comments.


Review 3

Summary and Contributions: The paper proposes an unsupervised domain alignment method based on log-likelihood ratio with normalizing flows. The method is designed to find a transformation of the dataset such that it it equivalent to another dataset with respect to a certain family of density functions. The main contributions of the work are a log-likelihood ratio minimizing distance metric based on normalizing flows with convergence guarantees. The optimization for unsupervised domain alignment is thus a minimization problem in contrast to adversarial formulations based on GANs. Experiments are performed on datasets -- the moon dataset and MNSIT-USPS.

Strengths: + The theoretical framework for minimizing the log-likelihood ratio for two distributions using normalizing flows is sound. + The method as shown in Figure 2 (last column) seems to have stable training (to convergence) compared to SN-GAN counterparts for domain alignment. + The qualitative examples are well structured to explain the advantages of using the proposed LRMF over the MMD, SN-GAN and AlignFlow.

Weaknesses: - Central parts of the paper are unclear eg. in line 80 \log P_M (X; \theta) should be the negative cross entropy. - The proposed objective Eq. 2 in line 128, requires the optimisation over both the parameters of the transformation \phi and the shared model \theta_S. The effect on the number of parameters vs. prior work eg. AlignFlow (Grover et. al. 2019) has not been discussed clearly. - The paper is sparse in quantitative results and does not compare to important prior work based on GANs [1]. The only quantitative results are on adaptation from USPS to MNIST in line 268. However, prior work [1] achieves 96.5% accuracy in comparison to the 55% accuracy achieved by the proposed method. - The empirical evaluation is restricted to small datasets eg. moons, MNIST and USPS. It would be desirable to evaluate the proposed approach on the more complex Facades/Maps/Cityscapes using the MSE metric to facilitate comparison with AlignFlow and [1]. - The shared model (\theta_s) is trained on two datasets simultaneously. It is unclear how the inductive bias from each of the datasets influence the shared space. [1] CyCADA: Cycle-Consistent Adversarial Domain Adaptation, ICML 2018.

Correctness: The empirical evaluation is limited to toy datasets whereas competing methods show better experimental setups (AlignFlow, CyCADA) and therefore improvement over the previous approaches is not established.

Clarity: -The paper can be improved with respect to introduction of technical notations. The definitions of (negative) cross-entropy can be made more consistent. - The introduction has various claims without appropriate citations. e.g, line 24 "difficult to quantitatively reason about the performance of such methods." line 31 " but rarely on density alignment". line 72 "In general, this would require solving an adversarial optimization problem" - line 191 "Authors of the AlignFlow" - >"AlignFlow"

Relation to Prior Work: The prior work discussion is limited to the work of SN-GAN, AlignFLOW, MMD. Various approaches for domain alignment based on GANs e.g. [1] are not discussed. The related work section can be improved.

Reproducibility: Yes

Additional Feedback: ---- Update after rebuttal-------- After the discussion and the rebuttal, I agree with the other reviewers that the idea is sound and has certain advantages over the prior work (AlignFlow). The scalability issue can be addressed in future work. However, the experimental section could have been improved as pointed to in the review above. I have modified my score accordingly.

[Author Response · NeurIPS 2020]

We would like to thank reviewers for their time and the effort they put into reviewing our submission. We welcome their thoughtful suggestions, which we will incorporate into the final improved version of this submission.

**Q1: (R1,3,4) Real datasets are too simple. A1:** We argue that small and controlled experiments provide more insight into the inner workings and failure modes of novel objectives. We are the first to introduce a purely likelihood-based alignment objective, and we empirically verify that resulting models preserve local structure of aligned domains better than non-parametric models, are free from mode collapse and training instability of GANs, and theoretically justify why they suffer from vanishing generator gradients in higher dimensions as much as their adversarial counterparts. We argue that this empirically verified result adds value to the scientific community, freeing future researchers from having to re-discover this confusing failure mode (vanishing gradients in higher dimensions) of a seemingly proper likelihood-ratio based alignment objective. Such findings are better explained with simpler, controlled experiments. We will add results on higher resolution images in other domains.

**Q2: (R4) No comparison to / improvement upon existing GAN-based domain adaptation methods in terms of classification accuracy. A2:** Recent improvements in the performance of classifiers adapted using adversarial alignment largely rely on additional losses utilizing source labels, such as semantic consistency in CyCADA [3] or classifier discrepancy in [4], added on top of the unsupervised adversarial alignment. In this work, we are primarily interested in improving the robustness of the alignment itself without making any assumptions about the downstream task. We will extend the related work section with recent advances in adaptation of classification models and explicitly acknowledge that the comparison to methods that make active use of source labels is beyond the scope of this work.

**Q3: (R1) Local optima of LRMF wrt Th 2.3 (3), when does the equality hold? A3:** The inequality (3) holds for any $\theta$ (shared model parameters) and $\phi$ (transformation parameters), but is tight (equality holds) only when the bias term is zero, and the shared model is optimal. However, the existence of local minima for theta might indeed prevent LRMF from converging to the actual value of LR-distance, *i.e.* the statement of the theorem always holds, but the final model produces a useful alignment only when a sufficiently "deep" minimum is found, otherwise the method fails, and the loss value is indicative of this failure. We will explicitly acknowledge this in the paper.

**Q4. (R4) Claims without proper citations. A4:** "difficult to quantitatively reason about the performance of [GANs]" - we will add [1; 2; 5]; "but rarely on density alignment" - we will cite AlignFlow and PointFlow here, as the only published attempts at doing that, to our knowledge; "In general, this would require solving an adversarial optimization problem" - we will append with "We show in Section 2 (Eq 1) that, in general, this would . . .".

**Q6: (R3) Extended description of results and setup on real data. How were transformation hyperparameters chosen?** We choose M that gives high validation likelihood on A and B separately, since optimal $\theta_S$ has to approximate A well. The choice of Real NVP vs GLOW is dictated by the same principle. We also show that in lower dimensions LMRF works even in the over-parameterized regime. We will extend the description of the setup and the discussion of results obtained on real data with an illustrated version of the following observation: in higher dimensions, more complex shared models quickly "envelop" two datasets before they become aligned, forming two disjoint "bubbles" of density that fail to merge (objective landscape is flat wrt $\phi$), whereas simpler shared models result in the alignment of means and variances only.

**Q7: (R4) Effect on the number of parameters vs AlignFlow. A8:** Indeed, compared to the AlignFlow-inspired baseline, our approach requires 2-3 times more parameters, but preserves the local structure of aligned domains better, as shown in the paper. Higher number of trainable parameters does not cause overfitting, as shown in Setup 1. However, GPU memory becomes an issue when training complex LRMF flows. We were able to fit a GLOW-LRMF with three scales and eight affine coupling layers on each scale with batch size = 4. This should be enough for medium-sized images from natural domains. We will explicitly acknowledge this in the related work and limitation sections.

**Q8: (R4) Inductive bias from two datasets in the shared model. A9:** Upon convergence the shared model has to fit the target dataset. It will be biased towards same kinds of patterns as the underlying flow model on target data.

**Q9. (R1,3,4) Clarifications and corrections:** *1) negative cross-entropy in line 80* - correct, we apologize for this typo, we double-checked, all other uses of this term in the paper are consistent with the literature; *2) $F \circ G^{-1}(A)$ composition refers to an AlignFlow-inspired baseline?* - correct; *3) should this be likelihood ratio than log-likelihood ratio?* - technically, you are right, but the term "log-likelihood ratio test" seems to be well adopted in the literature, though, technically, incorrect, since this is a "log of the likelihood ratio".

# References

[1] Shane Barratt and Rishi Sharma. A note on the inception score. *arXiv preprint arXiv:1801.01973*, 2018.

[2] Ali Borji. Pros and cons of gan evaluation measures. *Computer Vision and Image Understanding*, 2019.

[3] Judy Hoffman, Eric Tzeng, Taesung Park, Jun-Yan Zhu, Phillip Isola, Kate Saenko, Alexei A. Efros, and Trevor Darrell. CyCADA: Cycle consistent adversarial domain adaptation. In *ICML*, 2018.

[4] Kuniaki Saito, Kohei Watanabe, Yoshitaka Ushiku, and Tatsuya Harada. Maximum classifier discrepancy for unsupervised domain adaptation. In *CVPR*, 2018.

[5] Lucas Theis, Aäron van den Oord, and Matthias Bethge. A note on the evaluation of generative models. *arXiv preprint arXiv:1511.01844*, 2015.


[Meta-Review · NeurIPS 2020]

After discussion, all reviewers, and the meta-reviewer, agree that the paper should be accepted. As the authors show, the method in its current form may not scale well to higher dimensions. While a method without this limitation would obviously be preferable, the reviewers agree that this limitation can be addressed in future work, where the connection with GANs that the authors establish may be helpful.